

# Plasma Density Estimation from Ionograms and Geophysical Parameters with Deep Learning

Kian Sartipzadeh[1,2], Andreas Kvammen[3], Björn Gustavsson[1], Njål Gulbrandsen[3], Magnar G. Johnsen[3], Devin Huyghebaert[4], and Juha Vierinen[1]

[1]Department of Physics and Technology, UiT The Arctic University of Norway
[2]Simplera DA, Tromsø, Norway
[3]Tromsø Geophysical Observatory, UiT The Arctic University of Norway
[4]Leibniz Institute of Atmospheric Physics, Kühlungsborn, Germany

**Correspondence:** Andreas Kvammen (andreas.kvammen@uit.no)

**Abstract.** Accurate estimates of the ionospheric electron density are essential for various space-weather applications but are especially challenging at high latitudes due to strong spatial and temporal variability driven by auroral precipitation and complex ionospheric convection. This study presents an assimilative empirical model designed to improve regional electron-density estimates in Northern Scandinavia. The model uses ionogram images, the local magnetic field, the auroral electrojet, the

ring current and solar-activity indices as inputs. These inputs are fused by a multimodal neural network and trained with incoherent-scatter-radar (ISR) observations of electron density profiles as the target. The model remains functional with only a subset of input, leading to a modest accuracy degradation. Comparative analysis demonstrates that our neural-network–based assimilative model outperforms the ARTIST 4.5 ionogram scaler and the state-of-the-art E-CHAIM model, especially during auroral activity. Overall, our model achieves an $R^2$ score of 0.74 on an independent test dataset, whereas ARTIST 4.5 and

E-CHAIM obtain $R^2$ values of –0.08 and 0.34, respectively. These results indicate that the model can provide reliable, continuous electron-density estimates at high latitudes, even under auroral conditions. This methodology can be extended to develop empirical ionospheric models for other regions with historical ISR data and to invert ionograms to electron-density profiles when ISR observations are unavailable. A similar approach could also be applied in short-term forecasting of the ionospheric electron density.

## 1  Introduction

The Arctic ionosphere exhibits highly variable electron density due to auroral precipitation, structured plasma convection, and large seasonal variability in daily photoionization (Keskinen and Ossakow, 1983). Accurate knowledge of electron density profiles is important for a wide range of space-weather applications and ionospheric research (Buonsanto and Fuller-Rowell, 1997; Lanzerotti, 2001; Pulkkinen, 2007). Several measurement techniques allow continuous monitoring of ionospheric electron density, including GNSS total electron content (TEC) (e.g., Rideout and Coster, 2006) and ionosondes (Galkin et al., 2018). These

methods are less accurate than incoherent scatter radar (ISR) measurements, which observe Thomson scatter from ionospheric electrons and allow estimation of electron density, plasma temperature, and ion velocity (Evans, 1969; Beynon and Williams,





1978). However, unlike GNSS TEC and ionosonde measurements, ISRs provide significantly sparser coverage in time and space, often making continuous monitoring of electron density intractable.

Several methods have been developed to make use of ISR observations to facilitate the estimation of the ionospheric electron density. For example, Holt et al. (2002) and Zhang et al. (2005) used a global network of ISRs to develop regional empirical electron-density models that successfully capture ionospheric climatology. They binned observations by altitude and time of day, then derived linear relationships between solar activity, geomagnetic activity, and electron density. The E-CHAIM model

(Themens et al., 2017, 2018, 2019) incorporates solar variability via the F10.7 cm flux and the Ionospheric Index (IG) (Minnis, 1955; Liu et al., 1983). E-CHAIM uses information from ionosondes, ISRs, and radio-occultation experiments to generate an empirical model, providing improved plasma-density profiles in the auroral and polar regions compared with the International Reference Ionosphere (IRI) model (Bilitza et al., 2022).

Ionosondes measure the group delay of HF radio waves between a radar antenna and the point of reflection in the ionosphere across a range of frequencies (e.g., Davies, 1990; Levis et al., 2010). Inferring the electron-density profile from these measurements is challenging because the inverse problem is nonlinear and ill-posed, as there are regions in the ionosphere that are not directly measured, such as the "valley region" between the E-region peak and the altitude where the F-region electron-density exceeds the peak E-region electron density, and the top-side ionosphere where the electron density decreases. Several methods

have been developed to estimate the electron densities based on ionosonde measurements such as POLAN (Titheridge, 1985; Šauli et al., 2007), ARTIST (Reinisch et al., 1983, 1992; Reinisch and Huang, 2001; Reinisch et al., 2005; Galkin and Reinisch, 2008), NeXTYZ (Zabotin et al., 2006) and the IGC method (Ankita and Ram, 2023, 2024). These automated techniques perform well at mid-latitudes during conditions where the ionosphere is relatively unstructured, but are known to perform poorly during auroral precipitation, sporadic E and spread F conditions. Nevertheless, ionosonde measurements still contain infor-

mation about the F-region scale height variation, the electron-density in the valley region between the E-region peak and the F-region, the presence of auroral precipitation in the form of complex E-region ionogram traces, or in the form of weak or non-existent traces indicating a presence of high E- and D-region ionization. This information is currently not used optimally as there is no simple analytic relationship that can be used.

Machine learning (ML) techniques can uncover complex relationships within data and have greatly impacted terrestrial weather prediction (Schultz et al., 2021). In space physics, ML has been applied to a wide range of tasks (Camporeale, 2019), notably to classify auroral images (Clausen and Nickisch, 2018; Kvammen et al., 2020; Nanjo et al., 2022), predict ground-induced currents (Wang et al., 2019; Pinto et al., 2022), forecast substorm onset (Sado et al., 2023), predict local and global geomagnetic activity (Wintoft and Lundstedt, 1999; Wintoft and Wik, 2018), model space-weather drivers (Lundstedt, 2005),

scale ionograms (Chen et al., 2018; Rao et al., 2022; Guo and Xiong, 2023; Kvammen et al., 2024; Sherstyukov et al., 2024; Liu et al., 2025), perform time-series prediction of total electron content (Liu et al., 2022), predict electron-density distributions





(Mao et al., 2024), and classify plasma regions in near-Earth space (Breuillard et al., 2020).

In this study, we present a novel approach for estimating electron-density profiles above the EISCAT Tromsø ISR facility.
The goal is to find an empirical relationship between ISR-measured electron-density profiles and our input data: geophysical
parameters and ionograms. We employ deep learning (DL) to approximate a mapping from 25 readily accessible geophysical
parameters and/or ionogram images to the corresponding ISR measurements of electron density profile. This approach circum-
vents the need for algorithmic true-height inversion of ionograms and may capture more subtle information contained within
ionograms when estimating electron density. Convolutional neural networks have already been shown to be powerful tools
for automatic ionogram scaling (e.g., Kvammen et al., 2024). Here, we adopt a similar approach for extracting features from
ionograms, but instead of estimating scaling parameters, we directly predict the electron-density profile from each ionogram.

The remainder of this paper is organized as follows: Section 2 describes the dataset and preprocessing steps. Section 3
outlines the deep neural network architecture and training procedure. Section 4 presents our results and benchmarks them
against established models. Section 5 discusses the implications of these findings and proposes directions for future work.
Finally, in Section 6 we draw some conclusions from this work.

## 2 Data Acquisition and Preprocessing

In this study, we used three distinct data types acquired from multiple sources:

1. Electron-density profiles, $n_e(z)$, obtained from the EISCAT UHF incoherent-scatter radar at Ramfjordmoen, Tromsø
   (Rishbeth and Williams, 1985)

2. Ionograms recorded by the co-located digisonde operated by the Tromsø Geophysical Observatory (TGO)

3. Geophysical parameters retrieved from online space-weather and observatory services.

The EISCAT UHF profiles served as the "ground truth" for supervised training of our deep neural network (DNN). For each day
with available EISCAT UHF measurements, we extracted the corresponding ionograms and geophysical parameters. Figure 1
shows the resulting data availability: between 2019 and 2022, only 363 of 1 452 days (25%) yielded suitable data, providing 9
040 of 139 392 possible hourly profiles (6.5%), since most days lacked full 24-hour coverage. Data prior to 2019 were reserved
as an independent test set, containing 1 177 sample pairs. Our neural-network outputs were evaluated against ionogram-derived
electron-density estimates from ARTIST 4.5 (Reinisch et al., 2005) and E-CHAIM's $n_e(z)$ model predictions (Themens et al.,
2017).

### 2.1 The EISCAT Tromsø Ultra High-Frequency Radar

The EISCAT facility at Ramfjordmoen in Tromsø is one of the few high-latitude ISR stations located in the auroral zone
(69.6° N, 19.2° E). The UHF radar consists of a 32-meter antenna and operates in the 930 MHz band with a peak power of



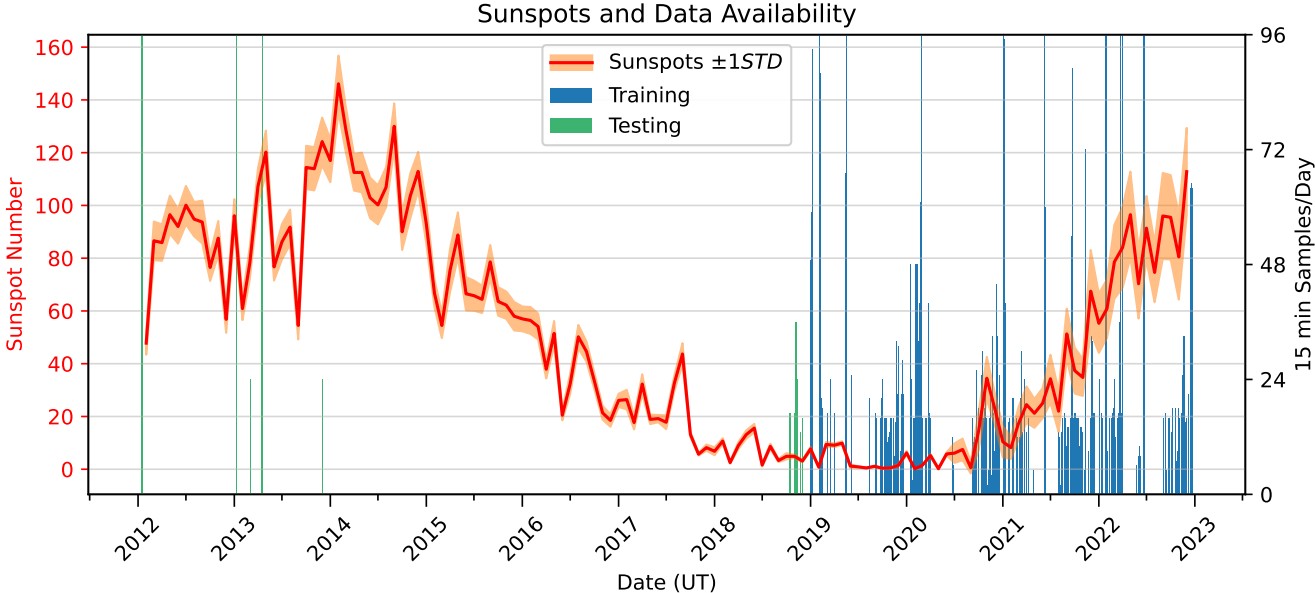

**Figure 1.** The distribution of 15 minute intervals with EISCAT UHF data suitable for this study are shown along with the Sunspot number (incinerating solar phase). Blue lines mark days that are used for training the model while green lines displays testing days.

2.0 MW. It measures ionospheric plasma parameters through several standard experiments (Lehtinen and Huuskonen, 1996). In this study, only data from runs of the standard experiment "beata" have been used. This experiment provides data with a

minimum time resolution of $5.0$ s over a range span of 49–693 km (Tjulin, 2017). One-minute-averaged plasma parameters for all available beata experiments, outlined in Figure 2, were extracted from the public Madrigal database (Madrigal Database, 2025) for further processing. Only the measurement time ($t$), radar range ($r$), electron density ($n_e$), and measurement error ($\sigma$) were processed. We discarded electron-density profiles measured at zenith angles exceeding $20°$ to focus on near-vertical scattering, thereby excluding most scanning experiments. Figure 2 (first column) illustrates the extracted profiles for a day with

full 24-h coverage.

The processed profiles were clipped to altitudes between 90 and 400 km to exclude D-region and upper F2-region activity, as measurements above 400 km contain little useful information for this study. The profiles were then re-interpolated onto the most common altitude grid in the dataset, consisting of 27 points. To remove NaN and $\pm\infty$ values (from failed ISR analyses),

we applied a simple in-painting procedure to interpolate or extrapolate these values. Additionally, a custom outlier-filtering script (see Appendix A) was used to detect and correct any remaining outliers in the $n_e$ profiles, which may arise from noise or calibration issues.

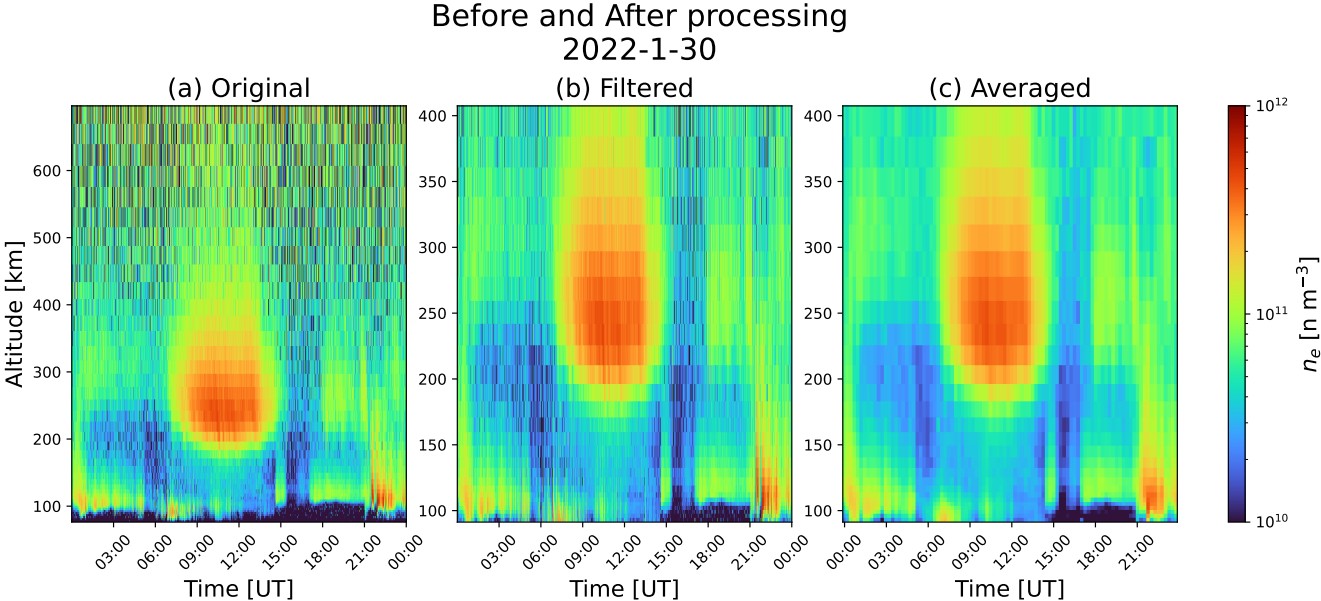

**Figure 2.** Example electron density profiles measured by the EISCAT UHF on 30 January 2022, illustrating the data processing steps. The left panel shows the original extracted profiles, including unclipped measurements with noise and outliers. The middle panel shows the profiles after outlier detection and filtering using PCA and IQR methods, followed by median filtering (kernel = 5). The right panel shows the final profiles after averaging over 15-minute intervals to match the temporal resolution of the Tromsø Digisonde.

Because the TGO ionosonde produces ionograms at least every 15 minutes, we averaged the EISCAT UHF profiles over 15-minute intervals to match the ionosonde's time resolution (see third column in Figure 2). Some intervals contained intermittent gaps; in those cases, we averaged all available measurements. Although the UHF data are averaged over a 15-minute interval, each ionogram sweep requires 7–8 minutes, with much shorter dwell times at individual frequencies. Thus, under highly dynamic conditions the ionosphere may significantly change between the lowest and highest frequencies within a single sweep, making direct comparison with EISCAT UHF measurements complicated.

## 2.2 The Tromsø Geophysical Observatory Ionosonde

Ionospheric soundings have been performed in Tromsø for nearly a century. Hosted by the Auroral Observatory, Sir Edward Appleton established himself with an ionosonde during the Second International Polar Year (1932–33). Blueprints of this instrument were given to the observatory, a new instrument was built, and permanent soundings began in 1935. Over the years, five different ionosondes have been operated in Tromsø, producing one of the longest continuous time series in the world (Hall and Hansen, 2003). In 1992, a Lowell Digisonde DPS system (Reinisch et al., 1992) was acquired, featuring autoscaling capability via the ARTIST software (Reinisch et al., 2005; Galkin et al., 2008). Because autoscaling could sometimes be unreliable, manual scaling of noontime ionograms was continued. The 1992 system underwent several software and hardware upgrades



in 1998, 1999, 2001, and 2005. From 2005 to 2022, ARTIST 4.5 (Reinisch et al., 2005) served as the automatic ionogram scaler. In December 2022, the system was upgraded to a Digisonde DPS-4D (Reinisch et al., 2008) and has since been running
ARTIST 5.0 (Galkin and Reinisch, 2008).

The TGO digisonde ($69.6°$ N, $19.2°$ E) is located approximately 525 m from the EISCAT UHF radar. The system operates in the upper medium-frequency (MF) to high-frequency (HF) bands with a range resolution of 5 km and a peak power of 300 W. The transmitter is connected to a crossed-rhombic antenna suspended by a $\sim$50 m tower, and reception is performed using four
crossed-loop antennas. The digisonde transmits HF pulses into the ionosphere to determine the frequencies, $f_s$, at which wave reflections occur. It sweeps from 1 to 16 MHz, averaging 16 pulses for each frequency, taking up to 7 minutes with a 50 kHz step size. Using standard interferometric techniques, the angle of arrival of the backscattered signal is used to separate vertical from oblique backscatter. Electron densities are then obtained from the O-mode reflection at each virtual height through the relationship: $f_s \approx 8.98\sqrt{n_e}$, where the plasma density $n_e$ is in m$^{-3}$ (Breit and Tuve, 1926; Davies, 1965).


SAO-Explorer, the standard Digisonde analysis tool, was used to extract ionogram data for further processing. We extracted frequencies, virtual ranges, amplitudes, polarizations, and zenith angles to construct ionogram images. Vertical echoes with polarization $\pm 90°$ were classified as O- and X-mode reflections, respectively. Using the extracted frequencies and virtual ranges, standard RGB images of size $113 \times 301 \times 3$ were constructed, assigning O-mode to the red channel, X-mode to the green
channel, and leaving the blue channel empty. Virtual ranges were clipped between 80 and 480 km to approximately match the altitude range of the ISR $n_e$ profiles described in Subsection 2.1. The upper altitude limit was set to 480 km as ionosondes measure virtual heights and not actual heights, thus, allowing for the increasing difference between virtual and actual heights when nearing the F region maximum. Frequencies above 9 MHz were discarded, as few ionograms during this phase of the solar cycle contain $f_{OF2}$ above that threshold. Finally, a resampling grid was applied to produce symmetric ionogram images
of shape $81 \times 81 \times 3$. Figure 3 illustrates an example before and after processing.

Furthermore, during processing we found that ionograms measured before 2015 contained substantial noise around the X- and O-mode echoes (see lower left panel of Figure 3). These noisy echoes, reminiscent of leakage or a disconnected Rx antenna, could degrade model performance because it does not reflect the distribution of electron-density profiles in the training
data. We therefore filtered these ionograms by discarding all pixels whose amplitude $A_{f,i}$ at frequency $f$ and image height $i$ is less than 75% of the maximum amplitude at that frequency. In other words, we retained only pixels satisfying the condition: $A_{f,i} \geq 0.75 \max_i (A_{f,i})$.

## 2.3 Geophysical Parameters

A curated dataset of geophysical parameters was collected to characterize solar and geomagnetic activity. These parameters
serve as inputs to the machine-learning network alongside ionogram data, facilitating the development of a baseline ionospheric model that is adjusted when an accompanying ionogram is available. This approach enables estimation of ionospheric plasma





**Figure 3.** Top row: The left panel shows the original ionogram from the TGO ionosonde, with O-mode echoes in red and X-mode echoes in green, plotted as a function of frequency (x-axis) and virtual altitude (y-axis). The right panel shows the resampled ionogram, remapped onto an $81 \times 81$ grid with altitudes clipped to 80–480 km and frequencies limited to below 9 MHz. Bottom row: The filtering process in three stages. The left panel presents the original ionogram with noise around the echoes. The middle panel highlights the detected noise in black. The right panel shows the cleaned ionogram, with echo strength indicated by color intensity.



densities even during intervals without ionogram observations, thereby enhancing the network's robustness and supporting continuous, long-term operation.

In total, 25 parameters were selected to capture solar, global, auroral and local geophysical conditions. Solar activity indices (R, F10.7, and Ly-$\alpha$), ring-current indices (SYM-D, SYM-H, ASY-D, and ASY-H), and the ap index were obtained from the OMNIWeb database (OMNI Database, 2025), while auroral electrojet indices (SMU, SML, and SME) were retrieved from SuperMAG (SuperMAG Database, 2025). Local magnetic-field parameters were calculated from magnetometer observations in Tromsø, Norway, provided by TGO (TGO Database, 2025). The magnetometer is located approximately 14 km northwest
of the co-located EISCAT ISR and TGO ionosonde. The full list of parameters is summarized in Table 1 and plotted in Figure 4. The correlation matrix (Figure 5) reveals that some indices are nearly perfectly correlated, for example, sunspot number, F10.7, and Ly-$\alpha$ irradiance have correlation coefficients above 0.9. Still, all parameters are retained, as the computational cost of including each is negligible.

The cyclic parameters, day-of-year, time-of-day, and solar zenith angle, were normalized to the range $[-1, 1]$. All other geophysical parameters were normalized using robust scaling. For each parameter $p_i$, the scaled value $\tilde{p}_i$ is given by

$$\tilde{p}_i = \frac{p_i - \text{median}(p_i)}{\text{IQR}(p_i)}, \tag{1}$$

where $\text{IQR}(p_i)$ and $\text{median}(p_i)$ are the interquartile range and median of $p_i$, computed over the dataset spanning one solar cycle (2012–2022).

**Table 1.** Solar and geophysical parameters used in the model

| Parameter | Description |
|---|---|
| R | Sunspot number, indicative of solar cycle phase |
| F10.7 | Solar radio flux at 10.7 cm, proxy for solar activity |
| Ly $\alpha$ | Solar ultraviolet emission at 121.6 nm, important for ionospheric ionization |
| ap | Geomagnetic disturbance index, reflecting global geomagnetic storm activity |
| SYM-H, SYM-D | Symmetric magnetic disturbance, proxy for ring current strength |
| ASY-H, ASY-D | Asymmetric magnetic disturbance, indicative of interhemispheric differences |
| SMU, SML, SME | Auroral electrojet indices representing magnetic disturbances in auroral zones |
| $B_X^{min}, B_Y^{min}, B_Z^{min}$ | Minimum disturbances in local magnetic field components over 15 min |
| $\bar{B}_X, \bar{B}_Y, \bar{B}_Z$ | Mean local magnetic field components, indicative of average auroral activity |
| $B_X^{max}, B_Y^{max}, B_Z^{max}$ | Maximum local magnetic field component, indicative of peak auroral activity |
| $\chi$ | Solar zenith angle, indicating solar elevation angle |
| $DOY_{sin}, DOY_{cos}$ | Day-of-year encoded cyclically, capturing Earth-Sun geometry |
| $TOD_{sin}, TOD_{cos}$ | Time-of-day encoded cyclically, representing diurnal variation |

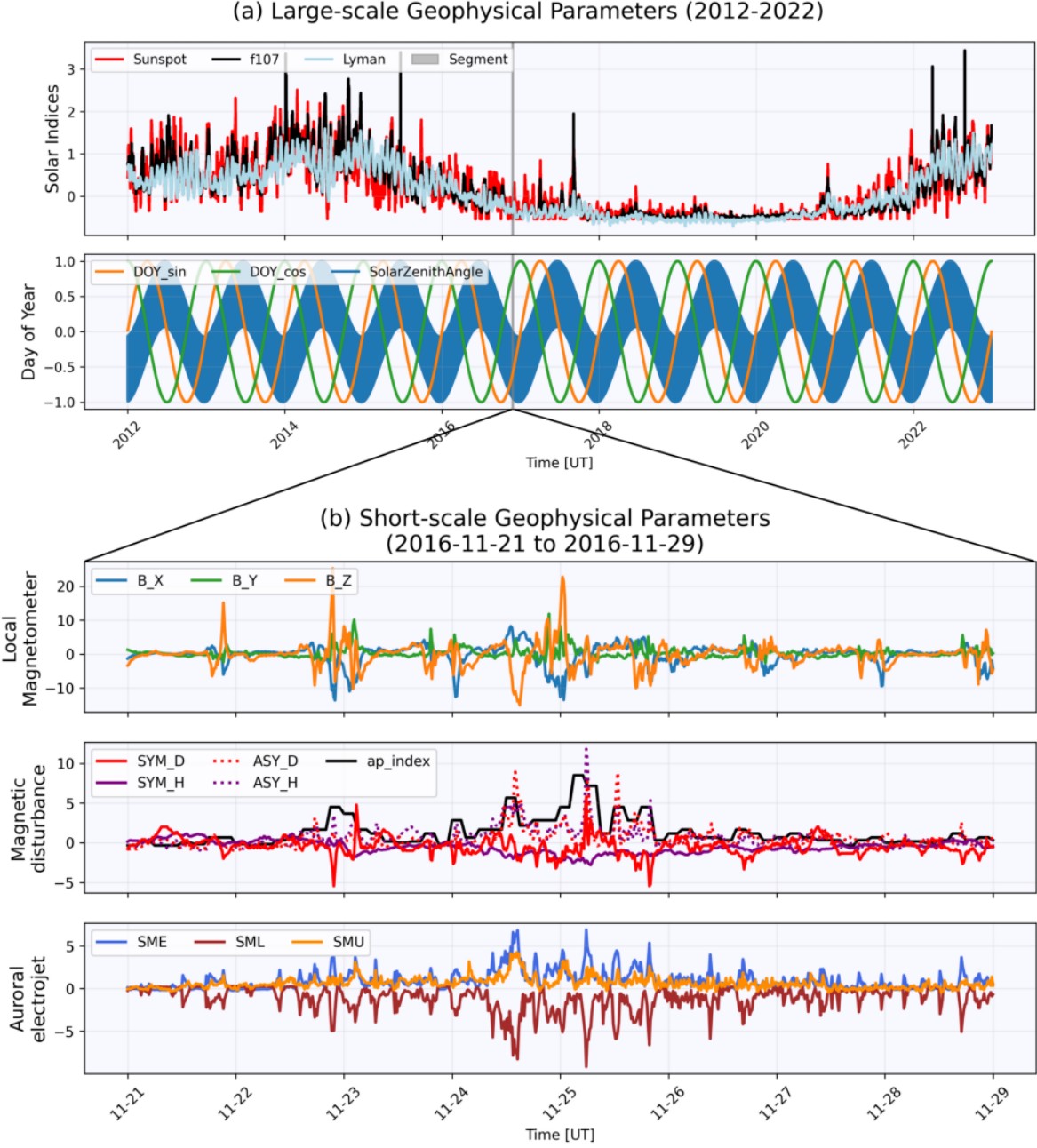

**Figure 4.** Time series of geophysical parameters. (a) Normalized large-scale solar parameters: sunspot number (red), F10.7 (black), and Lyman-$\alpha$ irradiance (light blue) and the day-of-year (DOY), plotted from 1 January 2012 to 31 December 2022. (b) A shorter interval showing additional normalized parameters: local magnetic-field components, ring-current indices, the Ap index, and auroral electrojet indices.





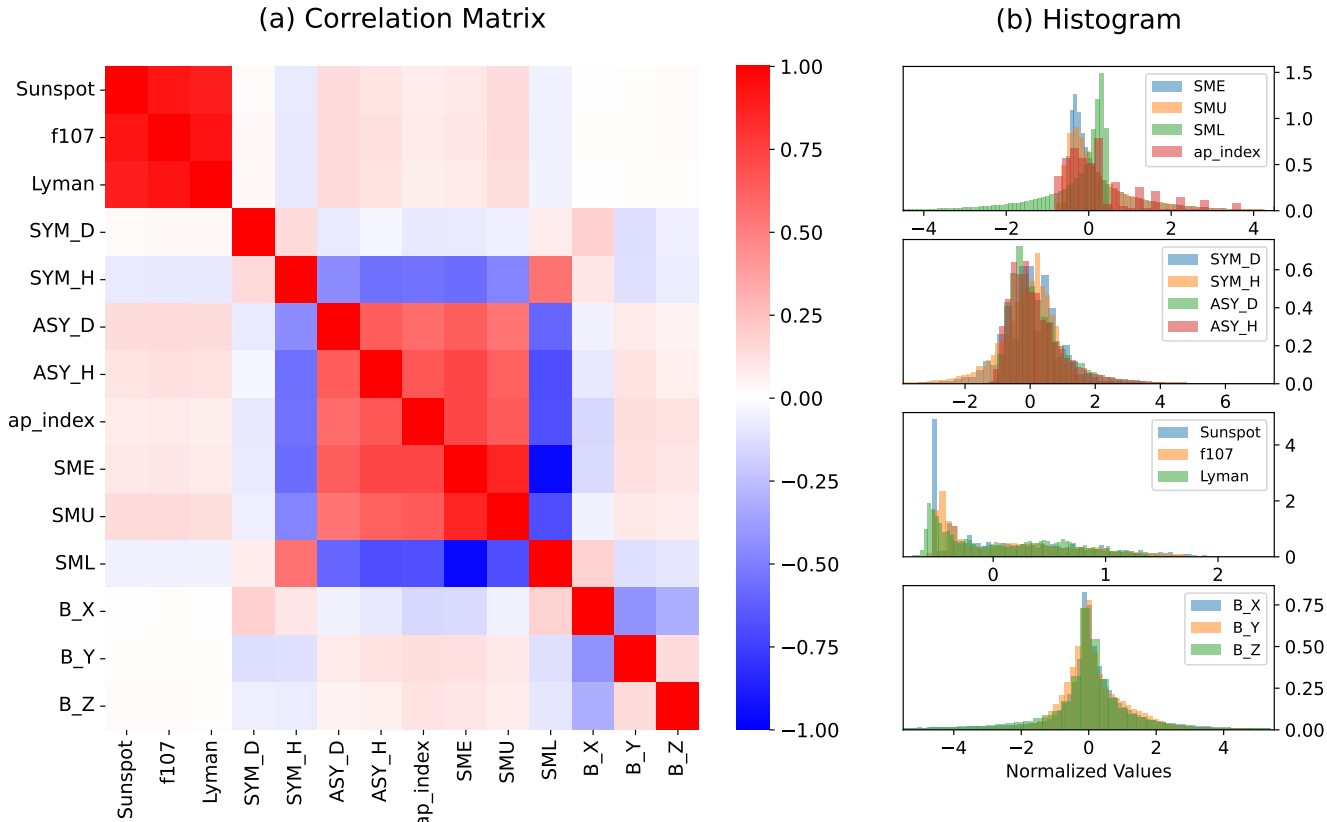

**Figure 5.** Overview of correlation and distribution of geophysical features from 2012 to 2022. (a) Correlation matrix of geophysical parameters excluding the time features and the min/max B-field features. (b) Histograms exhibiting the distribution for each feature category. i.e, Auroral electrojet, Magnetic disturbance, solar activity and B-field.

## 3 Deep Learning Methodology

This section presents the multi-modal neural network designed to estimate the electron-density profiles, $n_e(z)$, from ionograms and geophysical parameters. We describe the machine-learning and pattern-recognition techniques employed and detail the network architecture, which processes both scalar geophysical inputs and ionogram images.

### 3.1 Multi-Modal Neural Networks

The goal of this study is to estimate ionospheric electron-density profiles from corresponding ionogram images and geophysical parameters. We formulate this as a regression problem: find a function $f$ that takes an ionogram $I$ and parameter vector $p$ as inputs and produces the profile $n_e(z)$ as output. In practice, we model

$$n_e(z) = 10^{f(I,p)} \tag{2}$$



This defines a mapping between our input domains and the output domain:

$$f : (\mathbb{I}, \mathbb{P}) \rightarrow \mathbb{N}_e,\tag{3}$$

where $\mathbb{I}$ is the ionograms of size $H \times W \times C$ (here $81 \times 81 \times 2$ pixels), $\mathbb{P}$ is the $d = 25$ geophysical feature vector (see Section 2), and $\mathbb{N}_e$ is the electron-density profiles at the 27 considered height steps.

However, deriving the function $f$ is challenging both because the inverse problem is highly non-linear and because of the complex behaviour of the high-latitude ionosphere, especially during periods of auroral activity. Consequently, exploring machine learning, in particular neural networks (NNs), is appropriate for developing a mapping that captures these non-linear, complex relationships. A key property of NNs is their ability to approximate any non-linear function given sufficient hidden layers, neurons, and well-chosen hyperparameters (Goodfellow et al., 2016, Chapter 6). In this work, we take advantage of this property to develop a multi-modal neural network (MMNN) that combines subnetworks trained to extract relevant features from both ionograms and geophysical parameters. Thus, our regression is defined as:

$$\hat{n}_e(z) = 10^{f([h(I;\phi), g(p;\varphi)];\theta)},\tag{4}$$

where $h$ (with parameters $\phi$) processes the ionogram, $g$ (with parameters $\varphi$) processes the geophysical vector, and $f$ (with parameters $\theta$) combines their embeddings to predict $n_e(z)$.

This multi-modal configuration enables the MMNN to learn a richer latent representation by integrating different data sources, improving prediction accuracy (Huang et al., 2021). Moreover, the network can adapt to degraded or missing data in one modality by relying more heavily on the other. For example, if an ionogram is noisy or incomplete, the MMNN will predominantly base its density estimate on the geophysical inputs.

## 3.2 Network Architecture

Our MMNN consists of three networks: (1) a Convolutional Neural Network (CNN) (Goodfellow et al., 2016, Chapter 9) to extract spatial features from ionogram echoes, (2) a Deep Multi-Layer Perceptron (DMLP) to extract scalar features from the geophysical parameters, and (3) a fusion DMLP that learns from the concatenated features of the previous two networks. We refer to these networks as Iono-CNN, Geo-DMLP, and Fu-DMLP, respectively, to simplify notation. Together, they constitute KIAN-Net (Kian's Ionospheric Assimilative Neural Network). Figure 6 provides an overview of the KIAN-Net architecture.

### 3.2.1 Iono-CNN

CNNs are able to extract features from ionograms containing E- and F-region echoes, as shown by Kvammen et al. (2024). Although they used single-channel ionograms, their work inspired us to build a CNN for three-channel (RGB) ionograms. Our Iono-CNN (Figure 7) comprises four blocks, each containing:

- a Conv2D layer with $3 \times 3$ filters, stride 1, and padding 1;





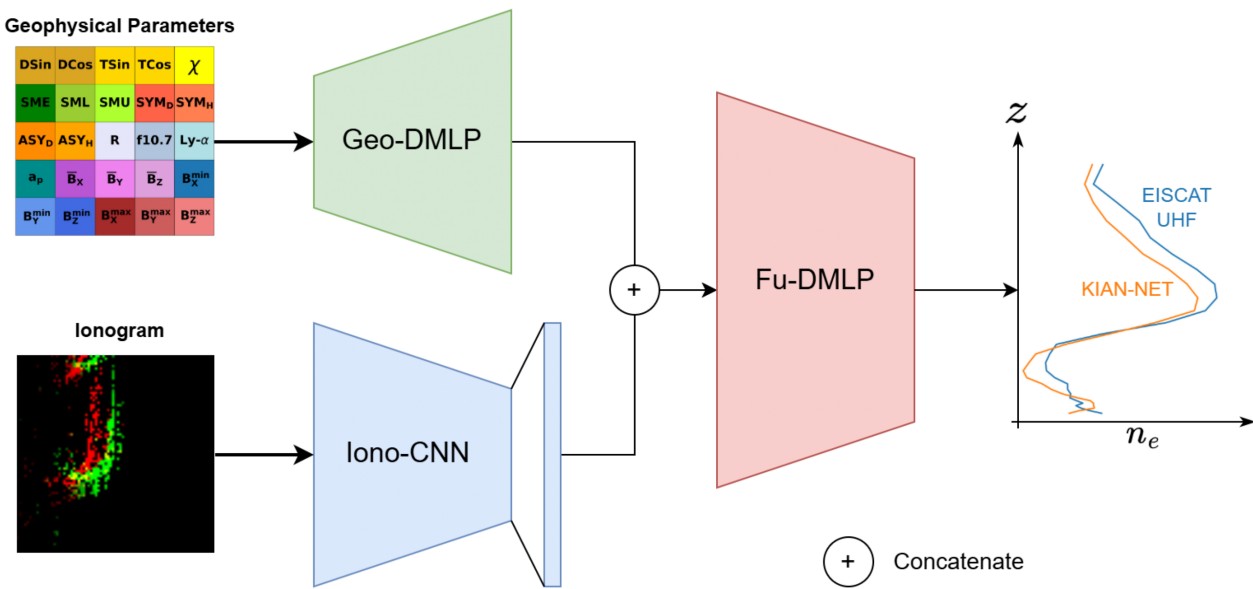

**Figure 6.** KIAN-Net architecture: A multi-modal neural network combining Iono-CNN for ionogram feature extraction, Geo-DMLP for geophysical parameter processing, and Fu-DMLP for feature fusion and final electron-density prediction.

 – a BatchNorm2D layer;

 – a ReLU activation (Nair and Hinton, 2010), which introduces nonlinearity while limiting outputs to $[0, \infty)$;

 – a MaxPool2D layer with a $2 \times 2$ window (except in the final block).

We initialize all weights with He (Kaiming) initialization (He et al., 2015) to preserve variance through the ReLU nonlinearity. In the final layer, the feature maps are flattened into a 12 800-dimensional vector. The BatchNorm2D layers both accelerate
convergence and smooth the loss surface by reducing internal covariate shift (Ioffe and Szegedy, 2015). Finally, the MaxPool2D layers downsample feature maps by selecting the maximum in each $2 \times 2$ window, the max-pooling lowers computational cost and captures spatially invariant dominant features (Goodfellow et al., 2016, Chapter 6).

### 3.2.2 Geo-DMLP

The Geo-DMLP maps 25 geophysical state parameters (Section 2.3) into a higher-dimensional (1024-dimensional) feature
vector for estimating electron density, $n_e(z)$. The network comprises five fully connected (FC) layers, each of which applies:

 – a fully connected layer with trainable weights and biases

 – batch normalization to reduce internal covariate shift (Ioffe and Szegedy, 2015)





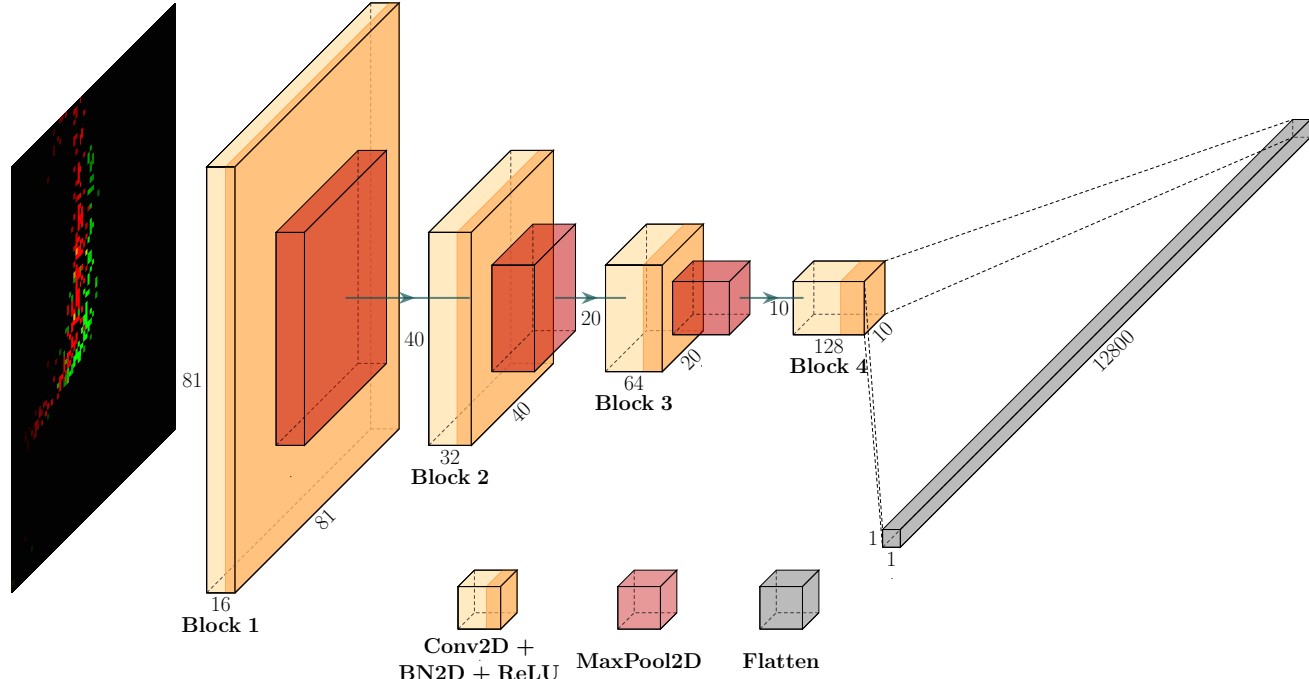

**Figure 7.** The Iono-CNN architecture: a 4-block convolutional neural network designed to extract spatial features from 3-channel (RGB) ionograms using stacked layers of Conv2D, BatchNorm2D, ReLU activation, and MaxPool2D. The diagram was made using the software developed by Iqbal (2018).

    – a ReLU activation (Goodfellow et al., 2016, Chapter 6) to introduce nonlinearity

The complete architecture is shown in Figure 8.

### 3.2.3 Fu-DMLP

To fuse the features from Iono-CNN and Geo-DMLP, we introduce Fu-DMLP (Figure 9). The network takes the concatenated feature vector (12 800 + 1 024 dimensions) and predicts 27 electron-density values across the 90–400 km range without explicit altitude inputs. Fu-DMLP is an eight-layer fully connected network arranged in an encoder-like fashion. In this model, each successive layer has approximately half the neurons of its predecessor, forming a bottleneck that compresses the representation before the final 27-dimensional output. Each layer consists of:

    – a fully connected layer with trainable weights and biases;

    – BatchNorm1D for stable activations (Ioffe and Szegedy, 2015);

    – a ReLU activation (Goodfellow et al., 2016, Chapter 6) to introduce nonlinearity.





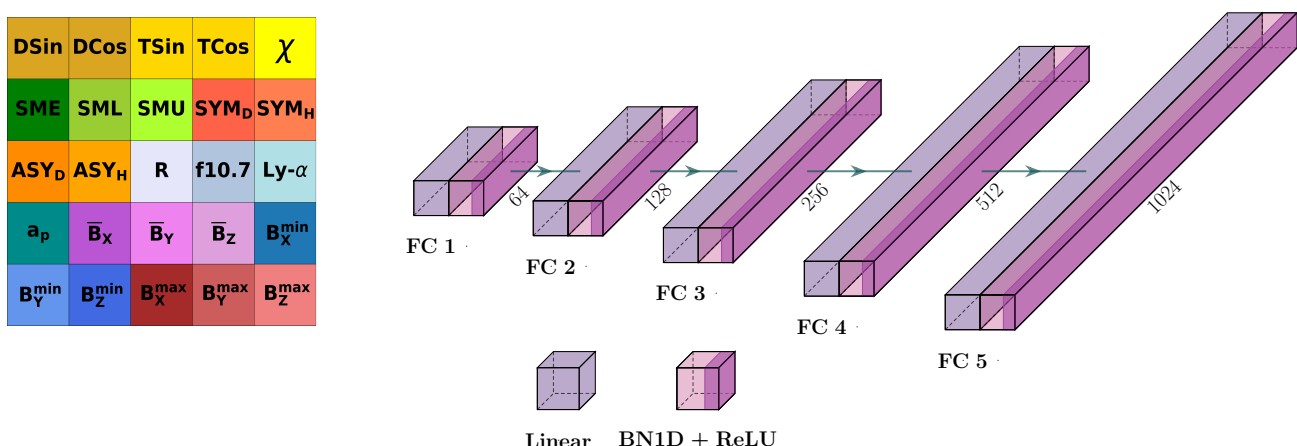

**Figure 8.** Geo-DMLP architecture: five fully connected layers transforming 25 geophysical inputs into a 1024-dimensional feature vector using FC layers, batch normalization, and ReLU activations. Inputs are color-coded by relevance (e.g., magnetic field components in shades of purple). The diagram was made using the software by Iqbal (2018).

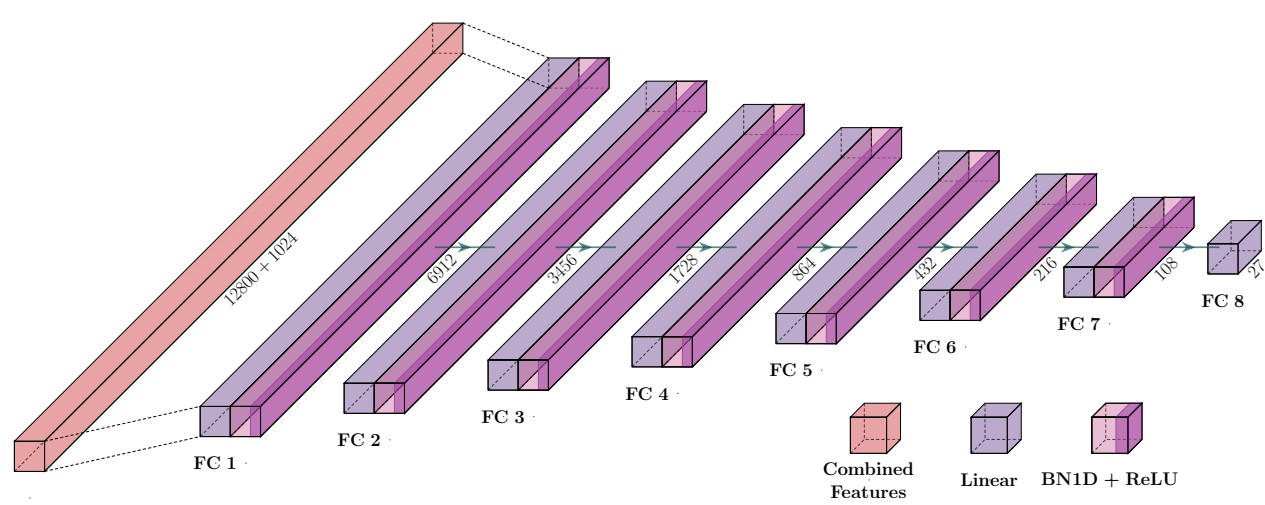

**Figure 9.** Fu-DMLP architecture: an 8-layer, encoder-style FC network that fuses the 12800 and 1024-dimensional feature vectors from Iono-CNN and Geo-DMLP to predict 27 electron-density values over the 90–400 km range. The model uses FC layers, BatchNorm1D, and ReLU activations for stable training and progressive dimensionality reduction. Diagram by Iqbal (2018).



**Table 2.** Architectural specifications of KIAN-Net's three-component MMNN. Iono-CNN processes 3-channel RGB ionograms through four convolutional blocks (Conv2D + BatchNorm + ReLU + MaxPool) with 3×3 kernels. Geo-DMLP trains geophysical parameters through five linear layers, also including batch normalization in 1D (BN1D). Fu-DMLP combines features from both networks and is trained to predict 27 electron density values. Parameter counts include BatchNorm weights within preceding layers. Total learnable parameters: $127,438,299$.

| Block | Layer Type | Output Shape | Specs (k, s, p) | Parameters |
|:---:|:---:|:---:|:---:|---:|
| | | **Iono-CNN** ($\phi$) | | |
| 1 | Conv2D + BN2D | (16, 81, 81) | (3, 1, 1) | 480 |
| | MaxPool2D | (16, 40, 40) | (2, 2, 0) | 0 |
| 2 | Conv2D + BN2D | (32, 40, 40) | (3, 1, 1) | 4,704 |
| | MaxPool2D | (32, 20, 20) | (2, 2, 0) | 0 |
| 3 | Conv2D + BN2D | (64, 20, 20) | (3, 1, 1) | 18,624 |
| | MaxPool2D | (64, 10, 10) | (2, 2, 0) | 0 |
| 5 | Conv2D + BN2D | (128, 10, 10) | (3, 1, 1) | 74,112 |
| | | **Geo-DMLP** ($\varphi$) | | |
| 1 | Linear + BN1D | (1, 64) | | 1,792 |
| 2 | Linear + BN1D | (1, 128) | | 8,576 |
| 3 | Linear + BN1D | (1, 256) | | 33,536 |
| 4 | Linear + BN1D | (1, 512) | | 132,608 |
| 5 | Linear + BN1D | (1, 1024) | | 527,360 |
| | | **Fu-DMLP** ($\theta$) | | |
| 1 | Linear + BN1D | (1, 6912) | | 95,572,224 |
| 2 | Linear + BN1D | (1, 3456) | | 23,898,240 |
| 3 | Linear + BN1D | (1, 1728) | | 5,977,152 |
| 4 | Linear + BN1D | (1, 864) | | 1,495,584 |
| 5 | Linear + BN1D | (1, 432) | | 374,544 |
| 6 | Linear + BN1D | (1, 216) | | 93,960 |
| 7 | Linear + BN1D | (1, 108) | | 23,652 |
| 8 | Linear | (1, 27) | | 2,943 |





### 3.3 Training

The predictions generated by KIAN-Net were evaluated against the ground truth using the Huber loss function (Huber, 1964).
This loss function combines the properties of both Mean Squared Error (MSE) and Mean Absolute Error (MAE), where small
residuals are treated quadratically, while large residuals are treated linearly. This makes the network more robust to outliers
(Terven et al., 2023), which can lead to more stable training that effectively damps the impact of short-scale ionospheric
activities. This approach further reduces the impact of noisy observations and faulty data on the network gradients. The electron

density profiles were normalized with $log_{10}$ in order to scale down in magnitude. Redefining the expression from Hastie et al.
(2009), Huber Loss is expressed as:

$$L_\delta(a) = \begin{cases} \frac{1}{2}(a)^2, & \text{if } |a| \leq \delta, \\ \delta|a| - \frac{1}{2}\delta^2, & \text{otherwise.} \end{cases} \tag{5}$$

where $a$ is the residual:

$$a = \log_{10}(n_e(z)) - f\left([h(I;\phi),\, g(p;\varphi)];\theta\right)$$

where $n_e(z)$ are the true values, $f$ are the model output values and $\delta > 0$ is a tunable hyperparameter that sets the transition
point between MSE and MAE. We used the default value $\delta = 1$. In order to estimate the mapping function in Equation 3, the
optimal set of total KIAN-Net parameters $\hat{\omega}$, where $[\hat{\theta}, \hat{\phi}, \hat{\varphi}] \in \hat{\omega}$, can be approximated by minimizing the total Huber Loss
over $N$ number of $n_e$ profiles. This is mathematically defined as:

$$\hat{\omega} = \underset{\theta,\phi,\varphi}{\arg\min} \sum_{i=1}^{N}\sum_{j=1}^{27} L_\delta\left(\log_{10}\left(n_e(z_j)\right) - f\left([h(I;\phi),\, g(p;\varphi)];\theta\right)\right). \tag{6}$$


Optimization was performed using the Adaptive Moment Estimation (ADAM) algorithm (Kingma and Ba, 2014), which
builds on the Stochastic Gradient Descent (SGD) algorithm, originally introduced by Robbins and Monro (1951). ADAM im-
proves SGD by using adaptive learning rates, estimating first and second order momentum terms with exponentially decaying

averages. These terms, mitigates the possibility of getting stuck in local minimas during training, as well as smoothening the
corresponding loss surface. However, it is acknowledged that recent research propose convergence issues linked to ADAM for
specific use cases (Reddi et al., 2019) (Zou et al., 2021).

Both the network architecture and the training process were implemented using the PyTorch framework. The dataset was
constructed by pairing ionograms and geophysical state parameters with the corresponding ground-truth electron-density pro-
files, forming sample triplets $(n_e(z)_i, I_i, p_i)$. This dataset was then randomly split into training and validation sets with an
80/20 ratio to mitigate splitting bias. Each set was loaded via PyTorch dataloaders: the training loader used a batch size of
and shuffled the samples after each epoch to reduce learning bias and improve generalization (Smith et al., 2020). For





validation, the entire validation set was utilized as a single batch, with the sample order also randomized. The training ran over
350 epochs with an initial learning rate set to 0.01. For refined convergence, two schedulers were set at 200 and 300 epochs,
reducing the learning rate to 0.001 and 0.0001, respectively.

## 4 Results

The global performance of KIAN-Net was evaluated by comparing its predictions $n_e(z_i)$ with EISCAT UHF measurements
from selected days in the years 2012, 2013, and 2018. The test dataset was entirely disjoint from the training dataset, using
different years of data for training and testing. This approach ensures a fair evaluation of the model's ability to generalize to
unseen geophysical conditions and solar cycle phases.

To determine how KIAN-Net's estimates of $n_e(z_i)$ compare with existing methods, we included results from ARTIST 4.5
and E-CHAIM in our analysis. The E-CHAIM model was run with both the "storm" and "precip" flags enabled, meaning
that global and auroral-zone information from the Ap, AE, and DST indices was used to estimate the electron density profile
(Themens et al., 2017). We also generated estimates using each network component separately (Iono-CNN and Geo-DMLP) to
assess their individual contributions and determine the relative influence of each input on the overall performance. To quantify
goodness of fit, we used the testing dataset to compute the $R^2$ score, the root-mean-square error (RMSE), the Pearson correla-
tion coefficient $r_P$, and the distance correlation $r_d$. A summary of these performance metrics are presented in Table 3.


**Table 3.** Comparison of five models using $R^2$-score, RMSE, Pearson correlation ($r_p$), and distance correlation ($r_d$) metrics, averaged over
all time steps in the testing dataset. KIAN-Net outperforms both its subcomponents (Iono-CNN and Geo-DMLP) and the baseline models
(Artist 4.5 and E-CHAIM) across all metrics.

| Model | $R^2$ | RMSE | r-pearson | r-dist |
|---|---|---|---|---|
| KIAN-Net | **0.74** | **0.13** | **0.92** | **0.92** |
| Iono-CNN | 0.66 | 0.14 | 0.89 | 0.90 |
| Geo-DMLP | 0.62 | 0.16 | 0.87 | 0.89 |
| ARTIST 4.5 | $-0.08$ | 0.77 | 0.55 | 0.60 |
| E-CHAIM | 0.34 | 0.20 | 0.78 | 0.83 |

To illustrate model performances and further analyze the model outputs, we examine a four-day interval (13–17 January
2012) that captures full diurnal cycles and a representative range of geophysical activity. Results for this period are presented
in Figure 10, with corresponding relative errors in Figure 11.





**Figure 10.** Comparison of electron-density profiles from 13 January 2012 to 17 January 2012 (00:00 UTC) against EISCAT UHF ground truth. (a) EISCAT UHF measurements of the electron density profiles. (b) Predictions made by KIAN-Net. (c) Predictions made by Iono-CNN. (d) Predictions made by Geo-DMLP. (e) Estimates made by ARTIST 4.5. (f) Estimates made by E-CHAIM. (g) The calculated $R^2$-scores at each time step where the colours depict the model with the best corresponding $R^2$.





**Figure 11.** Relative error between model predictions and EISCAT UHF ground truth from 13 January 2012 to 17 January 2012 (00:00 UTC). (a) Relative error by KIAN-Net. (b) Relative error by Iono-CNN. (c) Relative error by Geo-DMLP. (d) Relative error by ARTIST. (e) Relative error by E-CHAIM. (f) Colour plot depicting the colour of the prediction with the least absolute relative error. (g) RMSE at each time step, with colors indicating the model achieving the lowest RMSE.





Visual inspection of Figures 10 and 11 shows that all deep-learning models successfully approximate the electron-density profiles and agree well with the EISCAT UHF measurements. By direct comparison, KIAN-Net, Iono-CNN, and Geo-DMLP outperform ARTIST 4.5 and E-CHAIM on the test dataset, with superior quantitative metrics, as shown in Table 3. Overall, KIAN-Net, Iono-CNN, and Geo-DMLP achieve $R^2$ scores of 0.74, 0.66, and 0.62, respectively, while ARTIST 4.5 and E-CHAIM obtain $R^2$ of -0.08 and 0.34.


     Figure 12 presents the time-averaged performance metrics as a function of altitude. The metrics: $R^2$, RMSE, Pearson correlation ($r_p$), and distance correlation ($r_d$) are systematically better for the DL models at all altitudes below 350 km. In this region, ARTIST 4.5 often failed and was replaced by a constant $n_e = 1 \times 10^8 \, \mathrm{m}^{-3}$ to enable comparison with ISR observations. The performance gap is most pronounced in the E-region (100–160 km), where the KIAN-Net and Iono-CNN models achieve
$R^2 \gtrsim 0.4$ and $\mathrm{RMSE} \lesssim 0.2$, whereas ARTIST 4.5 and E-CHAIM yield $R^2 \ll 0$ and $\mathrm{RMSE} \gtrsim 0.2$. This is further illustrated by the relative-error plots in Figure 11(d) and (e), which show that densities between 90 and 160 km are generally underestimated (blue shading). Correlation metrics also favour the DL models, with both $r_p$ and $r_d$ typically exceeding 0.90, and reaching $\geq 0.95$ in the upper F-region for the KIAN-Net and Iono-CNN models, while ARTIST 4.5 and E-CHAIM exhibit correlations generally below 0.8.

**5   Discussion**

The estimates produced by KIAN-Net closely match those of Iono-CNN, indicating that the ionogram inputs dominate over the geophysical parameters used by Geo-DMLP. This is expected, since the ionogram traces contain direct information about the electron densities, whereas the relationship between electron densities and geophysical parameters is more indirect. Nevertheless, Geo-DMLP captures both the daily variation of the F-region and the auroral variation of the E-region $n_e$ relatively well.
At F-region heights, $n_e$ is governed by the balance between recombination and photoionization, which to first order depends on EUV flux (captured by F10.7) and the solar zenith angle. For the test days, Geo-DMLP tends to overestimate the duration of the F-region, predicting higher $n_e$ during both sunrise and sunset. We attribute this to significant day-to-day fluctuations of $n_e$ due to variable ionospheric, horizontal convection during these periods. Remarkably, Geo-DMLP also predicts E-region enhancements due to auroral precipitation, despite its limited information about the local ionosphere. The geophysical param-
eters most sensitive to auroral activity are the local magnetic-field fluctuations driven by the auroral electrojet. However, these fluctuations arise from currents spanning a much larger region than the localized E-region echoes detected by the ionogram. Further, our results show that the DL-models successfully captured nighttime electron density profiles with both E and F-region peaks.

ARTIST 4.5 systematically produced reasonable estimates of daytime bottom-side F-region densities. However, the quality of the topside densities is rather variable. This shortcoming stems from ARTIST's methodology, which derives an $\alpha$-Chapman profile for the topside F-region peak using bottomside F-region densities. This approach is an oversimplification, especially

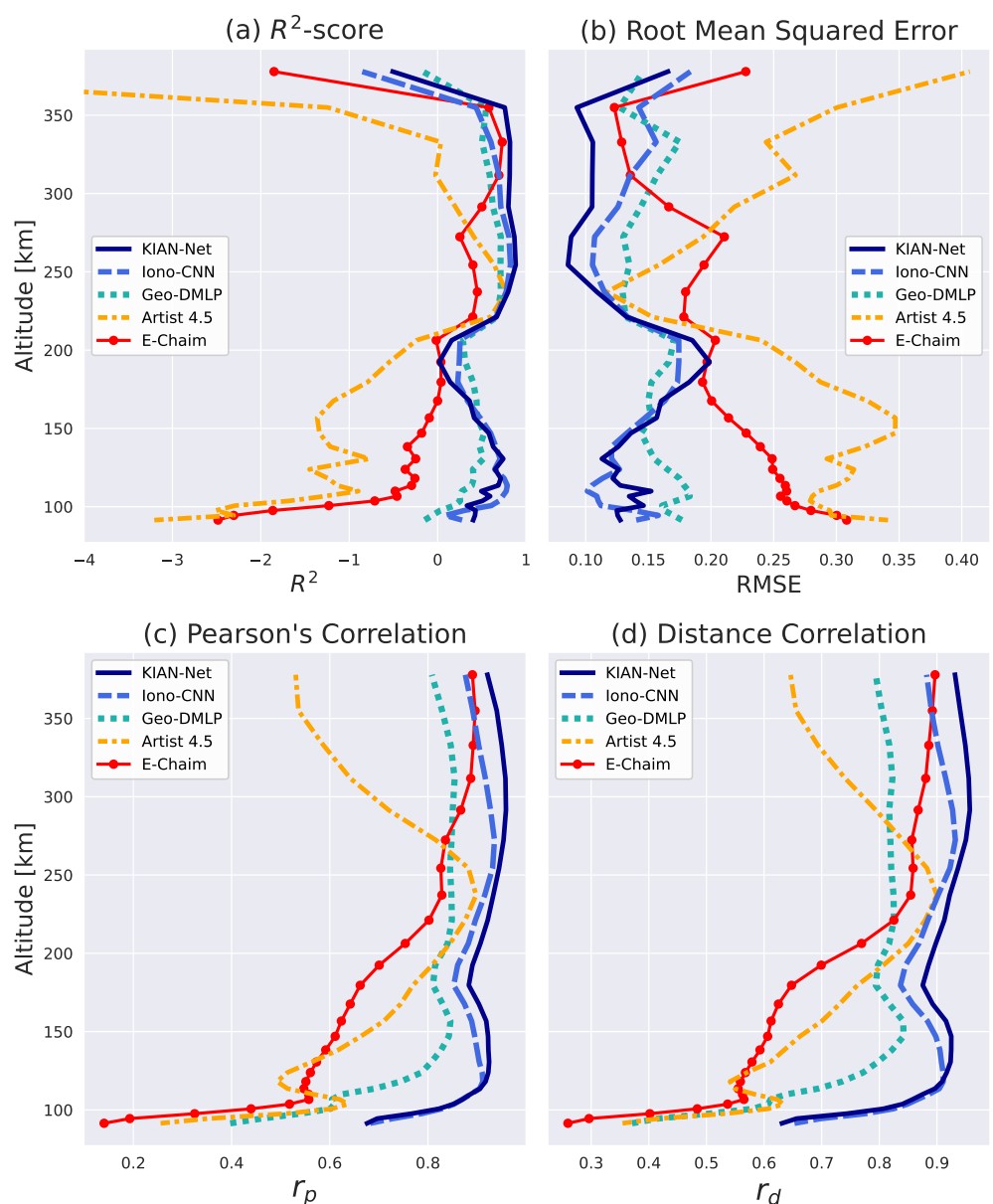

**Figure 12.** Calculated metrics between the models predictions (KIAN-Net, Iono-CNN, Geo-DMLP, ARTIST 4.5, E-CHAIM) and the EIS-CAT UHF electron density profiles along the temporal axis for each altitude ranging within 90-400 km. The performance metrics: (a) Calculated $R^2$-score, (b) Root Mean Squared Error (RMSE), (c) Pearson's correlation coefficient ($r_p$), and (d) The Distance correlation ($r_d$), are shown as functions of altitude. Results are aggregated over all available time steps where EISCAT UHF has valid profiles (non-Nan values), with metrics calculated using normalized (0-1) electron density values. Higher $R^2/r_p/r_d$ and lower RMSE indicate better agreement with EISCAT UHF measurements.





during periods of high-energy deposition from the magnetosphere and loss of thermal equilibrium along the vertical column. During auroral activity, ARTIST 4.5 often fails to produce any $n_e(z_i)$ estimates. When intense E-region ionization disrupts the
ionosonde's ability to detect F-region echoes, ARTIST omits to make an electron density profile and instead scales the E-region as a sporadic E-layer.

In comparison, E-CHAIM, as an empirical mid- and high-latitude ionospheric model, generated continuous predictions throughout the test period. Using only the Ap, AE, and Dst indices as inputs, E-CHAIM does not respond to short-term or local
variations, resulting in smoother $n_e(z)$ profiles. Consequently, E-CHAIM does not fully capture day-to-day local variations of F-region densities and the rapidly varying auroral E-region densities. Near the foF2 peak, where all models perform reasonably well, KIAN-Net and Iono-CNN remain slightly superior, with $R^2$ scores typically above $0.95$ and RMSE $\approx 0.05$. ARTIST 4.5 follows with $R^2 \approx 0.75$ and RMSE $\approx 0.13$, outperforming Geo-DMLP in this region. E-CHAIM shows weaker performance across most altitudes, except in the upper F-region ($\geq 250$ km), where its correlation coefficients reach $\gtrsim 0.65$.


There are several potential directions for future development of this method. First, incorporating additional EISCAT data from solar-maximum and summer conditions would balance the current dataset. Due to data availability, the considered dataset is weighted toward autumn, winter, and spring during the rising phase of the recent solar cycle. Dedicated summer campaigns should be conducted in the coming years to collect these missing observations. Second, extending this technique to include data
from other ISR facilities, both across magnetic local time in the northern auroral zone (Sondre Strømfjord, Resolute Bay, Poker Flat, and EISCAT Svalbard) and at lower latitudes (Millstone Hill, Sanya, Arecibo, and Jicamarca), would broaden geographic and temporal coverage. For this task, it is not obvious if a single deep learning model should be trained with all available data to provide a global ionogram-inversion scheme, or if it is preferable with separate models for each IS-radar followed by an interpolation scheme, such as those used in Holt et al. (2002) and Zhang et al. (2005). Although KIAN-Net can operate
using only ionograms or only geophysical parameters, we have not yet systematically evaluated how its performance degrades when one or more geophysical inputs are unavailable. Such "reduced-input" Geo-DMLP variants may effectively transition toward empirical models like E-CHAIM, this might make it possible to justify and make such models more responsive to geophysical conditions. Moreover, as computational resources increase, processing higher-resolution ionogram images with larger virtual-height ranges should further improve the quality of the estimates. Furthermore, E-CHAIM also includes an
assimilative mode, A-CHAIM, which ingests ionosonde and GNSS-TEC data to adjust its output away from climatology, comparing A-CHAIM's assimilative performance with KIAN-Net would be of interest. Likewise, incorporating vertical TEC as an additional geophysical input to KIAN-Net may enhance its accuracy. Finally, because ionograms may contain information about other plasma parameters, training KIAN-Net to predict electron and ion temperature profiles could extend its utility beyond density estimation.





## 6 Conclusions


We have presented a multi-modal neural-network, named KIAN-Net, to estimate ionospheric electron-density profiles from both ionogram images and geophysical parameters. The model was trained to learn the mapping from 25 readily available geophysical inputs and ionogram images to simultaneous electron-density profiles measured by the EISCAT UHF incoherent scatter radar in Tromsø. Its two subnetworks, Iono-CNN and Geo-DMLP, extract complementary features from the iono-

grams and geophysical state parameters, respectively. KIAN-Net and its two subnetworks performs favorably compared to the industry-standard ionogram-inversion algorithm (ARTIST 4.5) and the state-of-the-art high-latitude empirical ionospheric model (E-CHAIM). Notably, Geo-DMLP, which is trained only on geophysical parameters, effectively captures high-latitude spatial and temporal variability, caused by dynamic auroral precipitation and complex plasma convection, suggesting its utility as a local ionospheric model for Tromsø.


In total, over $10\,000$ ionogram–geophysical-parameter sample pairs were used to train and evaluate KIAN-Net. On an independent test set spanning multiple years, KIAN-Net achieved an $R^2$ of 0.74 against EISCAT observations, compared to $R^2 = -0.08$ for ARTIST 4.5 and $R^2 = 0.34$ for E-CHAIM. KIAN-Net also obtains $\mathrm{RMSE} \leq 0.17$, Pearson correlation $r_p \geq 0.92$, and distance correlation $r_d \geq 0.92$. Detailed analysis shows that KIAN-Net provides robust and reliable electron

density estimates ($R^2 > 0.5$) across diverse ionospheric conditions, including periods with significant auroral activity. Future work includes expanding the network to different latitudes and longitudes where ISR data are available and training it to predict additional plasma parameters (e.g., electron and ion temperatures) from the input data, further expanding the capabilities of the model.

*Code and data availability.* Ionograms and magnetometer data used to develop KIAN-Net were provided by TGO (TGO Database, 2025).

Ionogram data is also available trough the GIRO database (Reinisch and Galkin, 2011). Solar activity indices (sunspot number $R$, radio flux F10.7, and Lyman-$\alpha$ flux), ring current indices (SYM-D, SYM-H, ASY-D, and ASY-H), and the ap index were obtained from the OMNIWeb database (OMNI Database, 2025), while auroral electrojet indices (SMU, SML, and SME) were obtained from SuperMAG (SuperMAG Database, 2025). EISCAT electron density observations, used to evaluate the model's performance, are available from the Madrigal database (Madrigal Database, 2025). The replication code for this project is available at https://github.com/Kian0902/Masters_Project.

**Appendix A: Outlier-Filtering Procedure**

The outlier-filtering procedure operates as follows. First, a principal component analysis (PCA) reduces each electron-density profile to its first two principal components. Second, outliers are detected using the interquartile range (IQR) method, where any value lying outside the lower and upper fences at $\mathrm{Q1} - 1.5\,\mathrm{IQR}$ or $\mathrm{Q3} + 1.5\,\mathrm{IQR}$, where $\mathrm{IQR} = \mathrm{Q3} - \mathrm{Q1}$, is flagged as a potential outlier. Finally, each identified outlier profile is processed by applying a median filter with kernel size $k = 5$, which




smooths the deviation by matching neighboring values while preserving the overall profile shape. Figure A1 shows an example

of this filtering on EISCAT UHF profiles from 31 March 2022.

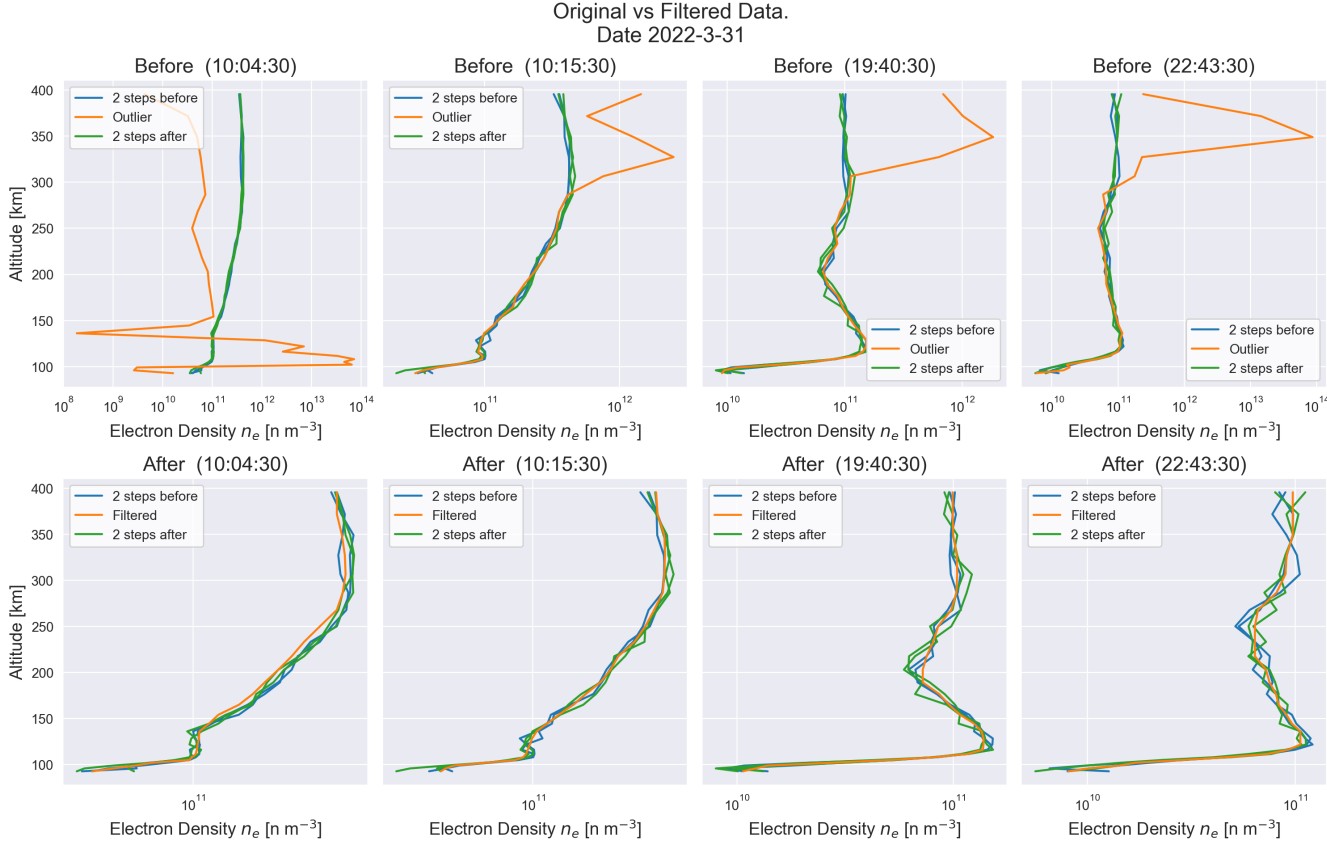

**Figure A1.** Example of outlier detection and median filtering applied to EISCAT UHF electron-density profiles from 31 March 2022. Each column corresponds to one detected outlier at different measurement times (labeled above each subplot). The top row ("Before") shows the original profiles, and the bottom row ("After") shows the profiles after median filtering ($k = 5$). The y-axis is radar range/altitude (km), and the x-axis is electron density on a logarithmic scale ($\mathrm{m}^{-3}$). The orange curve highlights the outlier profile, while the blue and green curves show the profiles from two time steps before and after the outlier, respectively.

## Appendix B: Statistical Comparisons of Peak Electron Densities

Given that ARTIST 4.5 demonstrates high accuracy in detecting F-region electron-density peaks, we implemented a peak-identification routine using the `find_peaks` function from SciPy's signal-processing library. Prominence and related param-

eters were tuned to capture as many true peaks as possible. E-region peaks were identified within 90–190 km and F-region peaks within 190–350 km. For each model output, the routine records the altitude and magnitude of each detected peak in




arrays corresponding to the original profiles. These model-derived peaks were then compared to those observed by the EIS-CAT UHF radar in a scatter plot (Figure B1). The performance at peak retrieval was quantified using $R^2$, RMSE, and Pearson correlation $r_p$ for each model output.


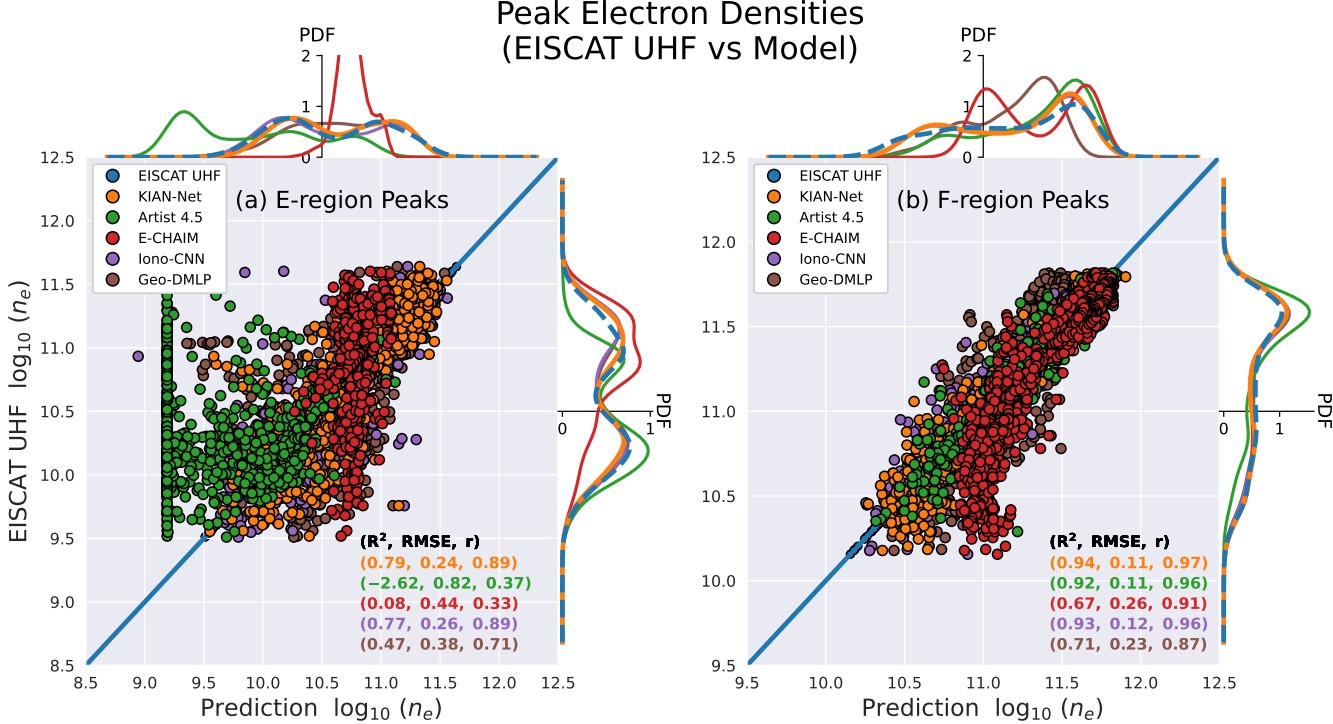

**Figure B1.** The E- and F-region peak electron densities measured by EISCAT UHF compared to peak predictions made by each prediction/estimation model. The $R^2$-scores together with the $RMSE$ and pearsons correlation $r$, are plotted in the same order corresponding to the legend. The PDFs are plotted respective to each axis to exhibit the variances in the distributions. (a) Detected E-region peak electron densities for each model plotted along x and y-axes (b) F-region peak electron densities for each model plotted along x and y-axes

To assess how well each model captures the overall distribution of peak electron densities, we estimated probability density functions (PDFs) for the E- and F-region peak magnitudes using kernel density estimation (KDE) with a Gaussian kernel (Koutroumbas and Theodoridis, 2008). Similarity between predicted and observed PDFs was quantified via the Bhattacharyya coefficient (BC) (Bhattacharyya, 1943), Jensen–Shannon divergence (JS) (Menéndez et al., 1997), and Wasserstein distance (W) (Villani, 2009), summarized in Table B1. These metrics were chosen to provide a multifaceted evaluation of the agreement between predicted and observed electron density distributions. The BC captures the amount of overlap between the two distributions, providing an intuitive measure of similarity. The JS-div quantifies the similarity between two probability distributions by symmetrizing the Kullback-Leibler divergence (Kullback, 1997), offering a bounded and smoother measure that






remains finite even when distributions do not share full support. The W-dist, reflects the minimal effort needed to transform
one distribution into the other, giving insight into the overall shape and displacement of the distributions.

**Table B1.** Model performance on E- and F-region peak predictions (Figure B1). Metrics for peak altitudes and magnitudes include $R^2$, RMSE, and Pearson correlation ($r_p$). Distribution similarity metrics for the PDFs are the Bhattacharyya coefficient ($BC$), Jensen–Shannon divergence ($JS$), and Wasserstein distance ($W$). Bold values indicate the best-performance.

| Model | E-region | | | | | | F-region | | | | | |
|---|---|---|---|---|---|---|---|---|---|---|---|---|
| | $R^2$ | RMSE | $r_p$ | $BC$ | $JS$ | $W$ | $R^2$ | RMSE | $r_p$ | $BC$ | $JS$ | $W$ |
| KIAN-Net | **0.79** | **0.24** | **0.89** | **0.99** | **0.03** | 0.06 | **0.94** | **0.11** | **0.97** | **0.99** | **0.06** | 0.12 |
| Iono-CNN | 0.77 | 0.26 | **0.89** | **0.99** | **0.03** | 0.06 | 0.93 | 0.12 | 0.96 | 0.98 | **0.06** | 0.11 |
| Geo-DMLP | 0.47 | 0.38 | 0.71 | **0.99** | 0.04 | **0.05** | 0.71 | 0.23 | 0.87 | 0.92 | 0.24 | 0.16 |
| ARTIST 4.5 | $-2.62$ | 0.82 | $-0.37$ | 0.78 | 1.75 | 0.64 | 0.92 | **0.11** | 0.96 | 0.97 | 0.09 | **0.10** |
| E-CHAIM | 0.08 | 0.44 | 0.33 | 0.62 | 1.19 | 0.37 | 0.67 | 0.26 | 0.91 | 0.93 | 0.23 | 0.12 |

For E-region peaks (90–190 km), both KIAN-Net ($R^2 = 0.80$, RMSE $= 0.23$, $r_p = 0.90$) and Iono-CNN ($R^2 = 0.81$, RMSE $= 0.24$, $r_p = 0.91$) demonstrate superior performance compared to the baseline models. As shown in Figure B1(a), Iono-CNN achieves the best peak-distribution alignment with EISCAT UHF peaks ($BC = 0.99$, $JS = 0.02$, $W = 0.05$), slightly outperforming KIAN-Net ($BC = 0.97$, $JS = 0.10$, $W = 0.12$). This indicates that direct ionogram inputs provide marginally higher 405 discriminative power for E-region peak prediction than the fused multi-modal approach. In contrast, ARTIST 4.5 produces mostly constant E-region peak densities around $\log_{10}(n_e) \approx 9.2$ and some varying clustering below $< 10.5$, resulting in poor metrics ($R^2 = -4.18$, RMSE $= 0.91$, $r_p = -0.17$) and low distribution similarity ($BC = 0.64$, $JS = 2.66$, $W = 0.82$). E-CHAIM similarly fails to capture E-region variability ($R^2 = -0.04$, RMSE $= 0.56$, $r_p = 0.06$), producing quasi-static peaks near $\log_{10}(n_e) \approx 10.7$ and resulting in a spiked PDF with low similarity ($BC = 0.25$, $JS = 2.97$, $W = 0.48$).

For F-region peaks (190–350 km), all models perform reasonably well. Iono-CNN maintains superior performance ($R^2 = 0.91$, RMSE $= 0.12$, $r_p = 0.96$), but ARTIST 4.5 slightly outperforms KIAN-Net in peak magnitude estimation ($R^2 = 0.89$, RMSE $= 0.11$ versus $R^2 = 0.87$, RMSE $= 0.14$), as summarized in Table B1 and visualized in Figure B1. This aligns with ARTIST's design focus on F-region Chapman profile modelling from foF2 information within ionograms, though its advantage 415 is restricted to peak magnitudes rather than full-profile accuracy. However, this advantage applies only to peak values and not to full-profile accuracy (Table 3). E-CHAIM shows reduced performance ($R^2 = 0.63$, RMSE $= 0.22$, $r_p = 0.81$) with a diverging tail forming slightly above $\log_{10}(n_e) \approx 11$. This is also observed in its F-region probability density function (Figure B), attributable to systematic underestimation during periods of enhanced ionization on 14 January 00:00 UT and 16 January 2012. This temporal misalignment is reflected in its elevated Wasserstein Distance ($W = 0.21$) compared to KIAN-Net ($W = 0.08$).



*Author contributions.* KS carried out the model development, training, and testing. KS created all figures and developed the project software with additions from AK. AK, BG, and JV conceptualized the initial study, and all authors provided additional ideas and suggestions as the project progressed. KS, AK, NG, and DH acquired the data used in this study. KS, AK, and BG wrote the main manuscript, with contributions from all authors. All authors contributed to the analysis of results and to reviewing and editing the paper.

*Competing interests.* The contact author has declared that none of the authors has any competing interests.

*Acknowledgements.* KS acknowledges support from the Space Physics Group at UiT – The Arctic University of Norway. AK, NG, and MGJ were funded by TGO – Tromsø Geophysical Observatory during this project. Norwegian participation in EISCAT is funded through the Research Council of Norway project 350179.



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
