# Peer review of "Plasma Density Estimation from Ionograms and Geophysical Parameters with Deep Learning"

_EGUsphere, 2025_

## Author Comment (AC1)

**General response to reviewers**

Dear reviewers,

Thank you for carefully reading through our manuscript and for providing appropriate comments. We have tried to address all issues.

Below is a comment-by-comment response to the reviewers' comments. The reviewer comments are in black, while the response is written with blue font.

with best regards,
Andreas Kvammen and co-authors

**Reviewer 1**

This manuscript presents a machine learning technique for reconstructing ionospheric profiles from ionograms and geophysical parameters, using incoherent scatter radar data as the truth data for training. The explanations in the paper are very clear, and the final results are convincing. I have a few minor comments for the authors to address.

**Minor comment 1.1.**

Use of ARTIST 4.5: The manuscript on line 120 says that the current processing for the Tromso ionosonde uses ARTIST 5.0 and ARTIST 4.5 has been discontinued. Why then do all the comparisons in this paper compare to ARTIST 4.5 not 5.0? Furthermore, this paper shows that ARTIST 4.5 performs very badly, to the point that it has a negative $R^2$ coefficient. What is making ARTIST 4.5 so bad in these comparisons? Line 293 says missing traces are being filled in with $1e8$ $m^{-3}$. Are those low placeholder numbers artificially wrecking the $R^2$ for ARTIST 4.5? I would expect ARTIST 4.5 to at least get the peak density correct since it should extract foF2 correctly.

These are good points that needs clarification in the manuscript:
1. ARTIST 5.0 has indeed been used on data acquired by the upgraded Digisonde DPS-4D since December 2022. However, bulk-processing data with ARTIST 5.0 is not straightforward, as this requires manual handling through SAO Explorer. To ensure sufficient data volume for training and evaluation, we therefore relied on the longer and more complete dataset from the previous Digisonde (2005-2022), which was scaled with ARTIST 4.5. To clarify this in the manuscript, we have revised the text (from line 122):

*In December 2022, the system was upgraded to a Digisonde DPS-4D (Reinisch et al., 2008) and has since been running ARTIST 5.0 (Galkin and Reinisch, 2008). For this work, however, we use data from the previous system (2005-2022), scaled with ARTIST 4.5. This allows us to take advantage of the large EISCAT dataset available from this period, providing the needed data volume to develop a reliable model.*

2. The reviewer is correct that the negative $R^2$ values are caused by the placeholder values used when traces are missing. With such a large fraction of

failed ionogram fits, it is difficult to summarize ARTIST 4.5's performance with a single statistical metric. On their own, these values do not provide a fair measure of its capability. Figures 10 and 11, however, provide additional context by visually illustrating the cases where the automatic scaling succeeds and fails. Indeed, ARTIST 4.5 generally captures the foF2 peak well when it attempts to scale it. This strength is highlighted and discussed in detail in Appendix B: Statistical Comparisons of Peak Electron Densities.

**Minor comment 1.2.**
Figure 10 panel g shows that at certain times, none of the techniques can reproduce the measured profiles and all R^2 drop to zero or lower. What is happening during these intervals where none of the techniques work?

That you for pointing out this important point. The low R^2 values appear primarily during nighttime conditions with weak auroral precipitation. To highlight this, we have added the following text to the results section (from line 294):

*Figure 10 also highlights that all models, including the deep learning models, struggle to estimate nighttime electron densities during weak auroral precipitation. There are several reasons for this. First, during low-density periods in the F region, such as under the ionospheric trough, ionograms often contain limited information about the electron density profile, particularly above auroral heights. The trough is typically associated with strong horizontal gradients and tilts in the ionosphere, which inhibit vertical-incidence signals from returning to the ionosonde receiver ((Voiculescu et al., 2010; Zabotin et al., 2006). This limitation can be further enhanced by auroral absorption in the D-region, which reduces the signal-to-noise ratio of the ionosonde returns. Second, there is also a differences in sampling geometry, the EISCAT UHF radar provides altitude-resolved measurements within a narrow 1° field of view, whereas the Digisonde has a broader vertical field of view 30°. This difference, combined with the ionosonde's reduced sensitivity during trough and absorption conditions, makes direct comparisons between EISCAT profiles and ionogram-based estimates particularly challenging in the nighttime auroral region.*

**Minor comment 1.3.**
Line 81 implies that only dates with a full 24 hours of coverage were used. Why is 24-hour coverage required if the model is being trained on individual profiles? This methodology appears to discard a lot of potentially usable training data.

Thank you for pointing this out, as the original wording may have been misleading. Full 24-hour coverage was not required, we also used data from days with partial coverage. This has now been clarified in the revised manuscript (line 78):

*Figure 1 shows the resulting data availability: between 2019 and 2022, 363 of 1 452 days (25%) yielded suitable data, providing 9 040 of 139 392 possible hourly profiles (6.5%), since many days contained data from experiments with partial daily coverage.*

**Minor comment 1.4.**

Why was 2019–2022 selected, and why was a larger portion of the available data not used for training? I am concerned about the biases in the training data, both including biases towards solar minimum and biases towards winter months.

We thank the reviewer for this valuable comment. The 2019–2022 period was selected due to data availability. The reduced coverage during summer months reflects user-community requests for lower system usage during this time. We acknowledge that this leads to seasonal imbalance in the dataset, and we plan to curate a more balanced dataset in future work. Regarding solar cycle coverage, our training dataset is indeed biased toward solar minimum and the subsequent rising phase. To mitigate this limitation, we deliberately selected our independent test dataset from the previous solar maximum. This allows us to evaluate model performance under different geophysical conditions, as shown in Figures 10 and 11. We have clarified this point in the manuscript (from line 80).

*The 2019–2022 period was selected due to data availability. Note that coverage is biased away from summer months due to reduced system usage during summer. This results in a dataset that is not balanced across all seasons. Furthermore, the training dataset is dominated by solar minimum and rising-phase conditions. To partially address this issue, we reserved data from the previous solar maximum and data from 2018 as an independent test set, containing 1 177 sample pairs. This enables evaluation of the model under contrasting solar cycle conditions. In future work, we aim to curate a more balanced dataset across both seasonal and solar-cycle variations to further improve model generalization.*

**Reviewer 2**

The work addresses an important challenge in ionospheric research by introducing a multimodal deep-learning approach (KIAN-Net) for estimating electron density profiles from ionogram images and geophysical parameters. The manuscript is well written, easy to follow and reads well. The methodology is sound, the analysis is thorough, and the comparison with established models (ARTIST 4.5, E-CHAIM) is both relevant and convincing. The figures are clear, and the results are significant for both operational and research contexts. The study's novelty lies in combining ionogram and geophysical data in a unified neural network framework and benchmarking its performance under auroral and disturbed conditions, where traditional scaling techniques struggle.

I have only a few suggestions and clarifications that could strengthen the paper.

**Minor comment 2.1.**

In the discussion, the authors note that ionograms dominate the predictions compared to geophysical parameters. This is an important point that could be supported quantitatively. I recommend performing (or at least discussing) a feature importance analysis, e.g., SHAP values or permutation importance, to

quantify the contribution of each geophysical parameter to the model output. Such an analysis would reinforce the justification for including the geophysical parameters and could reveal which indices are most relevant to the model.

We thank the reviewer for this valuable suggestion. A feature importance analysis is indeed helpful for supporting the conclusion that ionograms dominate the predictions compared to geophysical parameters. In our study, this relative contribution is already addressed by directly comparing the performance of three architectures: Geo-DMLP (geophysical parameters only), Iono-CNN (ionograms only), and KIAN-Net (fused). The results show that Iono-CNN significantly outperforms Geo-DMLP, while KIAN-Net produces results closely aligned with Iono-CNN, demonstrating that ionograms provide the dominant contribution to model performance.

Regarding methods such as SHAP or permutation importance, we note that their direct application to a multi-modal architecture like KIAN-Net is not straightforward. SHAP would require separate analyses of the CNN and MLP branches, with pixel-level explanations for ionograms that are difficult to interpret in this context. Similarly, permutation importance is well suited for tabular features but is not meaningful for high-dimensional image inputs like ionograms. Since the reviewer's concern primarily relates to the geophysical parameters, we instead report a preliminary absolute gradient impact analysis of these inputs, calculated from 768 randomly selected samples.

[Figure]

This analysis suggests that the solar zenith angle and F10.7 index have the largest impact, reflecting their relevance for F−region ionization., while the AE index and local magnetic−field variation strength show stronger contributions than the other indices, reflecting their sensitivity to auroral precipitation. These results indicate that the geophysical parameters provide physically meaningful, complementary information, though their overall contribution remains secondary to the ionogram inputs. A discussion of these points has been added to the revised manuscript (line 327).

*To further assess their role, we performed a preliminary absolute gradient impact analysis of the geophysical inputs. This analysis indicates that solar zenith angle and F10.7 contribute most strongly, followed by AE and local magnetic−field variations, consistent with their expected influence on F−region ionization and auroral precipitation. Although ionogram data dominate model performance, these results suggest that the geophysical parameters provide complementary and physically meaningful information.*

**Minor comment 2.2.**
Line 40: Among the referenced scaling techniques, the author should also include Autoscala (Pezzopane & Scotto, 2007).

Pezzopane, M., and C. Scotto (2007), Automatic scaling of critical frequency foF2 and MUF(3000)F2: A comparison between Autoscala and ARTIST 4.5 on Rome data, Radio Sci., 42, RS4003, doi:10.1029/2006RS003581.AQ

We agree that this is an appropriate and relevant reference. The suggested citation has been added to the revised manuscript.

**Minor comment 2.3.**
Appendix A: In the reviewer's opinion, the outlier detection and correction are not completely clear in their methodology.
- I think that PCA should be explained further for readers who are not familiar with typical preprocessing steps in machine learning. In the reviewer's opinion, a brief description of PCA would thus be helpful.

We thank the reviewer for this suggestion. To clarify the methodology, we have expanded Appendix A with a short description of Principal Component Analysis (PCA). The new text explains the purpose of PCA, how it works at a conceptual level, and why it is applied in our outlier detection procedure. The text (from line 395) now reads:

*The outlier−filtering procedure operates as follows. First, a principal component analysis (PCA) is applied to each electron−density profile. PCA is a dimensionality reduction technique that maps the original variables (electron densities at different altitudes) onto a new set of orthogonal axes, called principal components. These components are linear combinations of the original variables and are ordered so that the first component explains the largest variance in the dataset, the second the next largest, and so forth. By keeping only the first two components, each profile is represented in a compact two−dimensional form that captures its dominant variability, even though the components no longer correspond directly to the original physical variables (altitude and density). Second, outliers are detected using the interquartile range*

*(IQR) method, where any value lying outside the lower and upper fences at Q1 –
1.5IQR or Q3 + 1.5IQR with IQR = Q3 – Q1 is flagged as a potential outlier.
Finally, each identified outlier profile is processed by applying a 2D median filter
(in altitude and time) with kernel = (5x5), which smooths the outlier by
matching neighboring values while preserving the overall profile shape. Figure
A1 shows an example of this filtering on EISCAT UHF profiles from 31 March
2022.*

- The authors say that outliers are values falling outside the IQR range.
  What does value refer to? Is it an electron density value at a certain
  height? Is it the whole profile? Over which values were the IQR and median
  calculated?

By "value" we are referring to the dimensionally reduced profiles obtained after
applying PCA. Specifically, each profile was transformed into the two
corresponding dominant principal components, and these were then used for
the IQR method. In this context, any data point with a component value falling
below Q1 – 1.5 × IQR or above Q3 + 1.5 × IQR is considered an outlier
according to the interquartile range criterion. This should now be clear in the
updated appendix text. Below is a Figure that illustrates the methodology in
detail on one day of data:

[Figure]

- Outliers are removed by applying a median filter with a kernel size of 5.
  What does this mean? Is the median filter the same as that which was
  applied to smooth the ISR images shown in Figure 2b? Do you apply the
  median filter to a time series of electron density at a fixed height (rows of
  the image shown in Figure 2a)?

This part of the text was unclear, thank you for pointing this out. The Figure 2
text now reads:

*Example electron density profiles measured by the EISCAT UHF on 30 January 2022, illustrating the data processing steps. The left panel shows the original extracted profiles, including unclipped measurements with noise and outliers. The middle panel shows the profiles after outlier detection using PCA and IQR methods, followed by filtering of the detected outliers with a 2D median filter (in altitude and time) with kernel = (5x5). The right panel shows the final profiles after averaging over 15-minute intervals to match the temporal resolution of the Tromsø Digisonde.*

Also, in the Appendix text (lines 403–405) we now write:

*Finally, each identified outlier profile is processed by applying a 2D median filter (in altitude and time) with kernel = (5x5), which smooths the outlier by matching neighboring values while preserving the overall profile shape.*

- According to the methodology, outliers are removed by performing median filtering with a kernel size of 5. As a median (not mean) filter, the reviewer expects the filtered profile to match one of the values of the previous or following 2 profiles shown in Figure A1. Are the profiles shown in blue\green already filtered with PCA? Is the filtered one also gone through PCA?

If the filter was applied only in the time dimension, this would be expected. However, the applied filter is a 2D median filter with a kernel/window size of 5×5. This means that, for each data point, the median is computed not only from values at the same altitude but also across neighboring data points above and below that altitude. This explains why the filtered profile do not match the previous/following profiles. This detail was not specified clearly in the previous manuscript, but should be clear in the updated Appendix A text (lines 403–405):

*Finally, each identified outlier profile is processed by applying a 2D median filter (in altitude and time) with kernel = (5x5), which smooths the outlier by matching neighboring values while preserving the overall profile shape.*

Regarding the profiles shown in Figure A1, some of them are already filtered while others are not (they are processed sequentially in time). Importantly, regardless of whether a given profile is filtered, the algorithm always operates with a window size of 5. The use of a median filter, rather than a mean filter, was deliberately chosen in order to minimize the influence of unfiltered data with extreme values on the final profile.

All profiles both filtered and unfiltered, have gone through the PCA first to reduce the feature dimension, then passed to the IQR-method in order to detect the outliers. We hope that this is now clear in the updated manuscript Appendix A.